# Design and Optimization of MEMS Resonant Pressure Sensors with Wide Range and High Sensitivity Based on BP and NSGA-II

**DOI:** 10.3390/mi15040509

**Published:** 2024-04-10

**Authors:** Mingchen Lv, Pinghua Li, Jiaqi Miao, Qi Qiao, Ruimei Liang, Gaolin Li, Xuye Zhuang

**Affiliations:** College of Mechanical Engineering, Shandong University of Technology, Zibo 255000, China; w2764699195@163.com (M.L.); llipinghua@sdut.edu.cn (P.L.); 13581071927@163.com (J.M.); 15063939891@163.com (Q.Q.); lrm25790@163.com (R.L.); ligaolin@sdut.edu.cn (G.L.)

**Keywords:** resonant pressure sensor, MEMS, BP, NSGA-II, algorithmic optimization

## Abstract

With the continuous progress of aerospace, military technology, and marine development, the MEMS resonance pressure sensor puts forward the requirements of not only a wide range but also high sensitivity. However, traditional resonators are hardly compatible with both. In response, we propose a new sensor structure. By arranging the resonant beam and the sensitive diaphragm vertically in space, the new structure improves the rigidity of the diaphragm without changing the thickness of the diaphragm and achieves the purpose of increasing the range without affecting the sensitivity. To find the optimal structural parameters for the sensor sensitivity and range, and to prevent the effects of modal disturbances, we propose a multi-objective optimization design scheme based on the BP and NSGA-II algorithms. The optimization of the structure parameters not only improved the sensitivity but also increased the interference frequency to solve the issue of mode interference. The optimized structure achieves a sensitivity and range of 4.23 Hz/kPa and 1–10 MPa, respectively. Its linear influence factor is 38.07, significantly higher than that of most resonant pressure sensors. The structural and algorithmic optimizations proposed in this paper provide a new method for designing resonant pressure sensors compatible with a wide range and high sensitivity.

## 1. Introduction

As the global industry develops and advances, higher requirements are placed on pressure sensors for measurement accuracy, sensitivity, and range. For example, submarines need to accurately monitor depth positions in the 0–1000 m range. This requires highly accurate and sensitive pressure sensors with a range of 10 MPa. Resonant MEMS pressure sensors are the most accurate and stable pressure sensors available. They calculate the pressure by detecting the eigenfrequency change in the resonator, without analog-to-digital conversion, which makes signal acquisition and processing convenient. Device accuracy is mainly affected by the mechanical properties of a single-crystal silicon structure, strong anti-interference ability, and stable performance. Thus, it is very suitable for ocean exploration and other areas that require the accurate measurement of pressure in complex environments. Currently, researchers have conducted extensive studies on the range and sensitivity of resonant pressure sensors [1,2,3]. In the research on a wide range, Xiang C et al. [4] deployed silicon islands on the device layer to enhance the equivalent stiffness and structural stability of the pressure-sensitive diaphragm, achieving a measurement range of 0.2–7 MPa, sensitivity of 2.26 Hz/kPa, and a linear influence factor of 15.386. The linear influence factor means sensitivity × range, which is the product of the sensitivity and range. Yu J et al. [5,6] designed electromagnetic-excited sensors and electrostatically excited sensors, obtaining high measurement ranges of 0.11–30 MPa and 0.11–50 MPa, respectively, by increasing the thickness of the sensitive diaphragm. However, the sensitivity of these two sensors was only 0.43 Hz/kPa and 0.066 Hz/kPa, with linear influence factors of 12.83 and 3.3, respectively. Yan P et al. [7] proposed a resonator made of SOI and SOG materials, with a sensitivity of 11.89 Hz/kPa within a range of 0.1–1 MPa, and a linear influence factor of 10.7. Lu Y et al. [8] introduced a resonant pressure sensor with a measurement range of 1 MPa, featuring a sensitivity of 13.1 Hz/kPa and a linear influence factor of 13.1. These pressure sensors increase the diaphragm stiffness by either increasing the diaphragm thickness or reducing the diaphragm area to minimize diaphragm deformation at high pressures, ensuring measurement accuracy under high-pressure conditions. However, thicker diaphragms may reduce the conversion ratio, thereby sacrificing sensitivity to varying degrees. To achieve a higher sensitivity, Han X et al. [9] proposed a high-precision differential resonator. The lower thickness of the sensitive diaphragm (75 μm) resulted in a high sensitivity of 35.5 Hz/kPa, but the low stiffness of the diaphragm made the structure fragile. Significant deformation under high pressure led to a decreased linearity and accuracy, resulting in a range of only 0–0.2 MPa and a linear influence factor of 7.1. Y. Li et al. [10] proposed a high-sensitivity MEMS resonant differential pressure sensor based on bulk silicon, with a sensitivity of up to 143.17 Hz/kPa, but with a range of only 0.11 MPa and a linear sensing factor of 15.75. Additionally, Shi X et al. [11] and Lu Y et al. [12] proposed resonator structures with linear influence factors of 11.12 and 14.14, respectively, achieving a high sensitivity. Currently, traditional resonant pressure sensors are unable to meet the performance requirements of both a wide range and high sensitivity due to structural limitations. Pressure sensors compatible with both range and sensitivity not only enhance the measurement accuracy, expand the application scope, and improve the system performance but also reduce sensor application costs. Therefore, investigating resonators compatible with both range and sensitivity is of significant importance.

The working principle of traditional resonant pressure sensors, as shown in Figure 1, utilizes a secondary stress transfer island–diaphragm structure [10]. When external pressure acts on the sensitive diaphragm, the stress generated by the deformation of the diaphragm is transmitted through the silicon islands to the resonant beam, causing a change in the resonant beam frequency. However, the sensor can only reliably convert pressure changes into frequency changes when the deflection deformation of the sensitive diaphragm is small. Under high pressure, excessive deformation of the sensitive diaphragm leads to a decrease in linearity and accuracy [13], affecting the sensor’s range.

In response to the current shortcomings of resonator structures, this paper proposes a vertical resonator structure for resonant pressure sensors, and the structural principle is shown in Figure 1b. The new structure eliminates the influence of excessive diaphragm deformation on the linearity, providing feasibility for designing sensors compatible with a high sensitivity and wide range. Additionally, parasitic modes exist in resonant pressure sensors, and the proximity of parasitic mode frequencies to the operating mode frequency can lead to a decreased measurement accuracy. To eliminate the influence of mode interference on sensor performance in the new structure and improve two key indicators, namely the structure sensitivity and range, a multi-objective optimization design scheme for resonant pressure sensors based on Back Propagation Neural Network (BP) and Non-dominated Sorting Genetic Algorithm-II (NSGA-II) algorithms is proposed. The optimized structure avoids the influence of parasitic modes and achieves device performance with a high sensitivity and wide range. The new structure provides a conceptual approach for designing resonant pressure sensors with a wide range and high sensitivity. Furthermore, the multi-objective optimization scheme for sensitivity and range proposed for resonant pressure sensors can be applied to the multi-parameter optimization of other sensor structures.

## 2. Principle and Design of High-Sensitivity and Wide Range Resonator Structure

The principle of the vertical resonator structure, with the resonant beam perpendicular to the sensitive diaphragm, is illustrated in Figure 1b. When external pressure acts on the sensitive diaphragm, the diaphragm transmits the pressure to the resonant beam through the connecting beam, causing a change in the resonant beam frequency. Compared to the original structure, in the new structure, when the diaphragm is subjected to external forces, it not only receives support force from the four edges of the diaphragm but also obtains support force from the resonant beam. Additionally, the sensor adopts a secondary stress transfer island–diaphragm structure to suppress the energy coupling between the resonator and the sensitive diaphragm, reducing the impact of resonance on sensor accuracy, and thus improving the Q factor and accuracy. The resonant beam in the new structure suppresses the deformation of the diaphragm, improving the equivalent stiffness of the pressure-sensitive diaphragm and the overall stability of the structure under high pressure. This effectively addresses the problem of linear attenuation and accuracy reduction caused by the large deflection deformation of the sensitive diaphragm under high pressure. Based on the principle shown in Figure 1b, we designed the resonant pressure sensor shown in Figure 2.

The structure of the resonant pressure sensor is depicted in Figure 2. The resonator consists of an upper cover layer, a resonant layer, and a lower cover layer. The square hole in the middle of the structure serves as the pressure hole, allowing fluid to apply pressure to the surroundings through the square hole. The pressure hole consists of two thick walls (support surfaces) and two thin walls (sensitive diaphragms). The support surfaces prevent the overall deformation of the structure and fix the sensitive diaphragms, which are responsible for sensing external pressure. The stress on the diaphragm is transmitted to the resonant beam through silicon islands and connecting beams, causing a corresponding frequency change in the resonant beam. Excitation and detection electrodes are located on both sides of the resonant beam, forming capacitors C1 and C2, respectively, with the resonant beam. The sensor operates by exciting C1 and collecting the capacitance frequency signal through C2. The signal is transmitted to external circuits through wire holes, and external pressure is calculated using the frequency–stress relationship.

## 3. Mathematical Modeling and Parametric Analysis of Resonators

The stress and deformation of the new structure can be equivalent to the structure shown in Figure 3. When pressure acts on the sensitive diaphragm, the structure will undergo deformation from Figure 3a to Figure 3b. According to the principle of force equilibrium in the same direction, the pressure P acting on the sensitive diaphragm should be equal to the support force of the diaphragm. The support force on the diaphragm can be divided into two sources: one is the support force around the diaphragm, and the other is the support force provided by the resonator beam. The force acting on the diaphragm satisfies Equation (1).
(1)$${P=P}_{1}{\;+\;2P}_{2}$$

In Equation (1), P represents the external pressure on the diaphragm, P_1_ denotes the support force around the sensitive diaphragm, and P_2_ signifies the support force provided by a single resonator beam.

Furthermore, according to the deformation analysis depicted in Figure 3b, the sensitive diaphragm, connecting beam, and resonator beam satisfy relationship 2 in terms of deformation:(2)$$w_{d}=w_{c}+0.5w_{r}$$
where w_d_ is the deformation of the diaphragm, w_c_ is the deformation of the connecting beam, and w_r_ is the deformation of the resonant beam.

By combining the displacement equation and the stress equation, the structure satisfies relationship 3:(3)$$\left\{\begin{array}{c} {P=P}_{1}{\;+\;2P}_{2}\\ w_{d}{\;=w}_{c}+0.5w_{r} \end{array}\right.$$

To solve the unknowns in Equation (3), calculate the stretching deformation w_r_ of the resonator beam, and subsequently determine the axial stress in the resonator beam.

### 3.1. Analysis of the Connecting Beam

According to Newton’s third law, the forces acting on the connecting beam and the resonator beam are equal in magnitude and opposite in direction. Therefore, the magnitude of the forces acting on both ends of the connecting beam is P_2_. The deformation of the connecting beam satisfies the conditions for solving the cantilever beam deformation, and the deflection deformation is expressed by Equation (4):(4)$$w_{c}=\frac{P_{2}{L_{c}}^{3}}{{12\mathrm{EI}}_{c}}$$
where E, L_c_, and I_c_ are the modulus of elasticity of the silicon, the length of the connecting beam, and the moment of inertia of the connecting beam cross-section, respectively.

### 3.2. Analysis of the Diaphragm

The sensitive diaphragm is a rectangular diaphragm with four fixed sides, and its deformation under pressure P_1_ conforms to the theory of small deflection deformation of diaphragms, as shown in the deformation of the sensitive diaphragm in Figure 3. For a sensitive diaphragm with width 2a and length 2b, its deflection deformation can be expressed as Equation (5):(5)$$\frac{{{\mathrm{Eh}}_{d}}^{3}}{{12(1-v}^{2})}\left(\frac{\partial^{4}w}{\partial x^{4}}+2\frac{\partial^{4}w}{\partial x^{2}\partial y^{2}}+\frac{\partial^{4}w}{\partial y^{4}}\right)=\frac{P_{1}}{S}$$
where w_d_ is the deflection of the diaphragm, S is the sensitive diaphragm area, v and $h_{d}$ are the silicon Poisson’s ratio and the thickness of the diaphragm, respectively. The bending stiffness of the diaphragm can be expressed as
(6)$$D=\frac{{{\mathrm{Eh}}_{d}}^{3}}{12(1-v^{2})}$$

The boundary conditions for a rectangular diaphragm with fixed connections on all four sides are
(7)$$\left.\begin{array}{c} w_{x=-a,a}=0,{\left.\frac{\partial w}{\partial x}\right|}_{x=-a,a}=0\\ w_{y=-b,b}=0,{\left.\frac{\partial w}{\partial y}\right|}_{y=-b,b}=0 \end{array}\right\}$$

The deflection deformation of rectangular diaphragms with fixed connections on all four sides can be represented by a trigonometric function as Equation (8):(8)$$w_{d}\left(x,y\right)=\overset{\infty }{\underset{m=1,3,5\cdots }{\sum }}\overset{\infty }{\underset{m=1,3,5\cdots }{\sum }}C_{\mathrm{mn}}\left(1+\cos\frac{m\pi x}{a}\right)\left(1+\cos\frac{n\pi y}{b}\right)$$

Substituting Equations (7) and (8) into Equation (5), the expression for the diaphragm deflection can be obtained as Equation (9):(9)$$w_{d}=\frac{{4P}_{1}a^{4}\left(\left(1+\cos\frac{\pi x}{a}\right)\left(1+\cos\frac{\pi y}{b}\right)\right)}{{S\pi }^{4}D\left(3+2\frac{a^{2}}{b^{2}}+3\frac{a^{4}}{b^{4}}\right)}$$

### 3.3. Resonant Beam Analysis

The stress–strain deformation of the resonant beam, as shown in Figure 3, can be represented by Equation (10):(10)$$w_{r}=\frac{P_{2}l}{\mathrm{EA}}$$
where A, w_r_, and l are the cross-sectional area, elongation in length direction, and length of the resonant beam.

For a resonant beam without damping and driving effects, the formula for the resonant frequency is expressed as Equation (11):(11)$$\mathrm{EJ}\frac{\partial^{4}w\left(x,t\right)}{\partial x^{4}}{\;-\;P}_{2}\frac{\partial^{2}w\left(x,t\right)}{\partial x^{2}}{\;+\;\rho }_{m}\frac{\partial^{2}w\left(x,t\right)}{\partial t^{2}}=0$$
where J is the moment of inertia of the resonant beam cross-section. $\rho_{m}$ is the density of silicon.

The boundary conditions for the simply supported beam of length l at both ends are
(12)$${\left.w\right|}_{x=0,l}=0{\left.\frac{\partial w}{\partial x}\right|}_{x=0,l}=0$$

The solution of Equation (11) can be expressed as
(13)$$w=w\left(x,t\right)=w\left(x\right)\cos\omega t$$

Substituting Equation (13) into Equation (11) and solving, under the action of axial force P_2_, the frequency equation for the first-order free vibration in the width direction of the beam with both ends fixed can be expressed as Equation (14):(14)$$w\left(P_{2}\right)={\left(\frac{E(4.73{)}^{4}{b_{r}}^{2}}{{12l}^{4}\rho_{m}}+\frac{{\left(4.73\right)}^{4}\left(0.295\right)P_{2}}{{12l}^{2}\rho_{m}h_{r}b_{r}}\right)}^{\frac{1}{2}}$$

In Equation (14), l, h_r_, and b_r_ represent the length, width, and height of the resonant beam, respectively. 

The first-order resonant frequency of the resonant beam along the width direction under the action of axial force is
(15)$$f=\frac{\omega_{1}\left(P_{2}\right)}{2\pi }=\frac{{\left(\frac{{E(4.73)}^{4}{b_{r}}^{2}}{{12l}^{4}\rho_{m}}+\frac{{\left(4.73\right)}^{4}\left(0.295\right)P_{2}}{{12l}^{2}\rho_{m}h_{r}b_{r}}\right)}^{\frac{1}{2}}}{2\pi }$$

### 3.4. Overall Structural Analysis

The stress and deformation of the structure under the pressure shown in Figure 3 can be summarized using Equation (3). By substituting Equations (4), (9), and (10) into the unknown variables of Equation (3), the axial force acting on the resonant beam satisfies the relationship expressed in Equation (16):(16)$$\frac{{4P}_{1}a^{4}\left(\left(1+\cos\frac{\pi x}{a}\right)\left(1+\cos\frac{\pi y}{b}\right)\right)}{{S\pi }^{4}D\left(3+2\frac{a^{2}}{b^{2}}+3\frac{a^{4}}{b^{4}}\right)}=\frac{P_{2}{L_{c}}^{3}}{{12\mathrm{EI}}_{c}}+\frac{P_{2}l}{2\mathrm{EA}}$$

Using P_2_ to represent P_1_, the equation can be expressed as
(17)$$\frac{{4(P-2P}_{2}{)a}^{4}\left(\left(1+\cos\frac{\pi x}{a}\right)\left(1+\cos\frac{\pi y}{b}\right)\right)}{{S\pi }^{4}D\left(3+2\frac{a^{2}}{b^{2}}+3\frac{a^{4}}{b^{4}}\right)}=\frac{P_{2}{L_{c}}^{3}}{{12\mathrm{EI}}_{c}}+\frac{P_{2}l}{2\mathrm{EA}}$$

The deflection at the midpoint of the diaphragm can be represented as
(18)$$\frac{{16(P-2P}_{2}{)a}^{4}}{{S\pi }^{4}D\left(3+2\frac{a^{2}}{b^{2}}+3\frac{a^{4}}{b^{4}}\right)}=\frac{P_{2}{L_{c}}^{3}}{{12\mathrm{EI}}_{c}}+\frac{P_{2}l}{2\mathrm{EA}}$$

The deflection coefficient at the midpoint of the diaphragm under a uniformly distributed load is given by
(19)$$k=\frac{{192a}^{4}{(1-v}^{2})}{\pi^{4}{{\mathrm{Eh}}_{d}}^{3}\left(3+2\frac{a^{2}}{b^{2}}+3\frac{a^{4}}{b^{4}}\right)}$$

The axial force P_2_ in the resonant beam is given by
(20)$$\frac{{(P-2P}_{2})k}{S}=\frac{P_{2}{L_{c}}^{3}}{{12\mathrm{EI}}_{c}}+\frac{P_{2}l}{2\mathrm{EA}}$$
(21)$$P_{2}=\frac{\mathrm{kP}}{\frac{{\mathrm{SL}}_{c}^{3}}{{12\mathrm{EI}}_{c}}+\frac{\mathrm{Sl}}{2\mathrm{EA}}+2k}$$

Substituting P_2_ into Equation (15), the relationship between pressure P and frequency f is given by Equation (22):(22)$$f={\left(\frac{10.428{{\mathrm{Eb}}_{r}}^{2}}{\pi^{2}l^{4}\rho }+\frac{{\mathrm{SAL}}_{c}^{3}{+6\mathrm{SlI}}_{c}{+24\mathrm{EkAI}}_{c}}{3.9\pi^{2}l^{2}{\rho h}_{r}b_{r}{\mathrm{kPEAI}}_{c}}\right)}^{\frac{1}{2}}$$

Randomly selecting parameters such as the length, width, and height of the resonant beam, connecting beam, and sensitive diaphragm, we calculated the resonant frequency of the resonator under a pressure of 10 MPa using the proposed mathematical model. A comparison with data obtained from COMSOL 6.1 software revealed a maximum frequency difference of 2754 Hz and a maximum error of 1.96%, indicating the high accuracy of the proposed mathematical modeling. The calculated and comparison results are shown in Figure 4.

## 4. Pressure Sensor Design and Optimization

To make resonant pressure sensors (absolute pressure sensors) not only capable of achieving a specific measurement range of 1–10 Mpa but also characterized by a high sensitivity, we systematically varied the individual parameters and analyzed their effects on the sensor sensitivity and interference frequency for the resonator beam, connecting beam, and diaphragm. Based on the analysis results, we selected the sensor structure with the highest sensitivity within the specified range.

### 4.1. Resonant Beam Structural Analysis and Design

According to Equation (15), the relationship between the length l, width h_r_, and height b_r_ of the rectangular resonant beam and the initial frequency and sensitivity is shown in Figure 5. Here, when the length changes from 800 μm to 1500 μm, the resonant beam frequency decreases by 60% on average, and the sensitivity also decreases slightly. When the height changes from 40 μm to 50 μm, the resonant beam frequency remains basically unchanged, but the sensitivity decreases by 0.6 Hz/kPa on average, which is 20%. When the width is increased from 20 μm to 30 μm, the resonant beam frequency increases by 50% on average, while the sensitivity decreases by 50%. Reducing both the width and height can effectively improve the resonant beam sensitivity, but the close proximity of the width and height will lead to interference between the horizontal and vertical resonant modes, so there should be a certain difference between the height and width in the parameter selection. Compared with reducing the height, reducing the width of the resonant beam not only improves the higher sensitivity but also reduces the intrinsic frequency and increases the resonant beam amplitude. Therefore, a smaller width rather than height is chosen in the structure design. Based on the resonant beam parameters and resonant frequencies of the existing structures [10,11,14], a resonant beam structure with a length of 1200 μm, a width of 20 μm, and a height of 40 μm is selected with an intrinsic frequency of 120 kHz.

Preliminary structural models have been established, and dynamic analysis of the structure has been conducted using COMSOL. The first six mode shapes of the resonator structure are shown in Figure 6, while the first three mode shapes of the resonant beam are illustrated in Figure 7.

Compared to higher-order modes, lower-order modes have advantages such as larger resonance amplitudes, easier excitation and detection, and stronger interference resistance. Therefore, in the selection of operating modes for resonators, lower-order modes are typically preferred. For this reason, in this study, the first-order horizontal resonance mode of the resonant beam is chosen as the operating mode, which is the fifth-order resonance mode in the overall structure. The frequencies of the first four modes of the structure are concentrated in the range of 50–80 kHz, which is significantly different from the operating mode at 120 kHz and can be disregarded. As the frequency of the resonant beam increases under external pressure, the frequency difference between it and the sixth-order mode gradually decreases until they intersect. To prevent the intersection of the operating mode and the sixth-order mode (interfering mode) within the measurement range, which would lead to an increase in the measurement error, finite element analysis is utilized to analyze the structural parameters affecting the interfering mode and the operating mode. Based on the analysis results, the structure is optimized to obtain the optimal sensitivity structure without mode interference within a specific range (1–10 MPa).

### 4.2. Connecting Beam Structure Analysis and Design

The relationship between the interference frequency of the sensor and the parameters of the connecting beam is shown in Figure 8 (left). The sensitivity and parameters of the connecting beam are depicted in Figure 8 (right), where w_c_, L_c_, and h_c_ represent the width, length, and height of the connecting beam, respectively. When the width w_c_ is increased from 200 μm to 500 μm, the average sensitivity of the structure increases by approximately 3.5 Hz/kPa, which is about a 230% increase. Meanwhile, the average interference frequency decreases by around 60 kHz, representing about a 30% decrease. Increasing the width significantly improves the sensitivity of the structure, but also reduces the interference frequency and compresses the range of frequency variation in the resonant beam operation.

The connecting beam is located in the resonant layer along with the resonant beam, as shown in Figure 2, with a height of 40 μm. When the length L_c_ of the connecting beam increases from 1500 μm to 1700 μm, the average sensitivity of the structure decreases by 0.4 Hz/kPa, and the average interference frequency decreases by 40 kHz. Increasing the length leads to a decrease in both the sensitivity and interference modal frequency, thereby limiting the performance parameters of the structure. To achieve a higher sensitivity and interference frequency, the length should be minimized. The length of the connecting beam can be represented by the width of the diaphragm, as shown in Equation (23).
(23)$$L_{c}=2a+c$$
where L_c_ represents the length of the connecting beam, 2a is the width of the pressure-sensitive diaphragm, and c denotes the designed electrode placement distance, which is 200 μm.

### 4.3. Sensitive Diaphragm Structural Analysis and Design

The relationship between the diaphragm parameters and interference frequency and sensitivity is illustrated in Figure 9, where 2a, 2b, and h_d_, respectively, represent the width, height, and thickness of the diaphragm. The width 2a increases from 600 μm to 1100 μm, resulting in an average decrease of 70 kHz in the interference frequency, representing a 30% reduction. The sensitivity of the structure with a height of 800 μm is maximized when the width is 900 μm, while the sensitivity of the structure with a height of 1000 μm is maximized when the width is 1100 μm. As the height 2b increases from 600 μm to 1100 μm, there is no significant change in the interference frequency, but the sensitivity increases on average by 1.5 Hz/kPa, representing a 150% increase. When the thickness h_d_ increases from 50 μm to 95 μm, the interference frequency increases on average by 30 kHz, a 20% increase, while the sensitivity decreases on average by 3 Hz/kPa, representing a 75% reduction.

### 4.4. Resonator Structure Optimization

The pressure and frequency of the resonant structure exhibit a highly linear relationship; hence, the sensitivity s can be represented in terms of the operating frequency S at a full-scale pressure of 10 MPa, as shown in Equation (24):(24)$${S=f}_{0}\;+\;s\;\times \;r$$
where S is the full-scale frequency, f_0_ is the natural frequency of the structure, the value is 120 kHz, s is the sensitivity of the structure, and r is the range; since resonator sensitivity is usually measured in Hz/kPa, the value here is 1000.

The design of the structure should aim for both a high range and high sensitivity. Therefore, within the determined range of 10 MPa, adjusting the parameters to achieve a higher sensitivity is necessary, meaning a higher full-scale operational frequency should be pursued. Additionally, to prevent interference mode effects in this structure, the interference frequency should be higher than the operational frequency at the full scale of the sensor.

Therefore, in the design of the structure, the goal should be to obtain the maximum full-scale resonant frequency while ensuring that the interference frequency is greater than the full-scale operational frequency.

In the structural parameters, the parameters of the resonant beam are related to sensitivity but not to interference frequency. The relationship between the length of the connecting beam and the width of the diaphragm is given by Equation (23), indicating a linear relationship between the two. The height of the connecting beam is the same as that of the resonant beam, which is 40 μm. These parameters can be selected through single-objective optimization.

For the width of the connecting beam and the width, height, and thickness of the sensitive diaphragm, there is usually an inverse relationship between the interference frequency and the parameters chosen for the full-scale resonance frequency. When parameters leading to higher full-scale resonance frequencies are selected, the interference frequency tends to be lower, whereas when parameters leading to lower full-scale resonance frequencies are chosen, the interference frequency tends to be higher. This contradictory relationship implies that single-objective optimization methods cannot meet the requirements for designing the optimal structure. Consequently, exhaustive search methods become one of the most common approaches for selecting structures from multiple parameters. However, exhaustive methods are not only inefficient and costly but also may fail to find the global optimum when dealing with large search spaces, leading to being trapped in local optima. Therefore, it becomes challenging to obtain the optimal structure of the resonator. To address this issue, a multi-objective optimization design scheme for resonant pressure sensors based on the BP algorithm and NSGA-II algorithm is proposed, achieving the simultaneous optimization of range and sensitivity.

#### 4.4.1. BP- and NSGA II-Optimized Structures

The BP neural network [15] is widely utilized due to its simplicity, strong nonlinear mapping capability, and high operability. It serves as a black-box function to solve engineering optimization problems with complex input–output relationships, making it particularly suitable for fitting the unknown relationships between sensor structural parameters and performance. NSGA-I introduced a non-dominated sorting and elite strategy, while NSGA-II [16], based on NSGA-I, proposed congestion and crowding comparison operators, facilitating fast search and improving the convergence speed of optimization. It is often employed for handling multi-objective optimization problems.

In order to extract the parameter mapping relationship of the resonator and construct a high-precision BP network, 300 sets of data were randomly selected for the four parameters that simultaneously affect the interference frequency and operating frequency, namely the width of the connecting beam and the width, height, and thickness of the diaphragm. The corresponding operating frequency *S* and interference frequency F for each parameter set were calculated using the commercial software COMSOL. Among these, 260 sets of data were chosen as the training set and 40 sets as the validation set for training the BP neural network. Partial data are shown in Table 1.

Build a BP neural network with four inputs and two outputs, using diaphragm width 2a, diaphragm height 2b, diaphragm thickness h_d_, and connection beam width w_c_ as input independent variables, and interference frequency F and operating frequency S as output dependent variables. The function relationship fitted by BP is used as the objective function for NSGA II iteration optimization to obtain the Pareto optimal solution set with two outputs. The optimization framework of the structure is illustrated in Figure 10. Calculate the interference frequency and operating frequency corresponding to the structural parameters using finite element analysis and import them as training and testing sets into the BP training network. The weight model trained by BP is imported into NSGAII as the functional relationship equation, and the Pareto solution set after 1000 iterations of population iteration is taken as the optimal solution set. The solution most in line with the requirements is selected from the Pareto solution set as the optimal structure, and finite element analysis is conducted to validate the structure.

The optimization model constructed employs a three-layer neural network structure for BP training, comprising an input layer, a hidden layer, and an output layer. Each layer is associated with corresponding weights and thresholds. The input matrix and output matrix are illustrated in Equations (25) and (26), respectively.
(25)$$X=\left[{2a\;2b\;h}_{d}{\;w}_{c}\right]$$
(26)$$Y=\left[F\;S\right]$$

The three-layer neural network structure consists of four input parameters and two output parameters, with a hidden layer containing nine nodes. The specific formula is 27.
(27)n=2m+1
where n represents the number of hidden nodes, and m represents the number of input layer nodes.

In the BP training process, if there are significant differences in the range of the data, it can lead to large biases during weight updates, resulting in slow training. To improve the training speed of the BP network and mitigate the issues of gradient vanishing or exploding, all data are normalized using the normalization equation as shown in Equation (28). Subsequently, the data are de-normalized using Equation (29).
(28)$$X_{i}=2\times \frac{\left(x_{i}{-x}_{\min}\right)}{x_{\max}{-x}_{\min}}-1\left(i=1,2,\cdots ,n\right)$$
(29)$$Y_{i}=\frac{\left(y_{i}+1\right)\times \left(y_{\max}{-y}_{\min}\right)}{2}{+y}_{\min}\left(i=1,2,\cdots ,n\right)$$

For the iteration of weights in the BP network model, adjustments are made using the error backpropagation algorithm. When the mean squared error E between the fitted output data and the actual values exceeds the expected value for the current iteration, the error is propagated backward from the output layer, and the weights and thresholds are updated until the error gradually decreases to achieve the desired accuracy. The mean squared error is calculated using the formula shown in Equation (30).
(30)$$E=\frac{1}{n}\overset{n}{\underset{i=1}{\sum }}{\left(p_{i}{-r}_{i}\right)}^{2}$$
where p_i_ is the predicted value of the BP fitting, and r_i_ is the true value of this training.

The NSGA-II algorithm employs a strategy of fast non-dominated sorting and crowding distance to achieve good convergence in multi-objective optimization problems. It can quickly converge to the Pareto front with a high convergence, solution set diversity, and uniformity. Its workflow is illustrated in Figure 11.

An initial population *Q*_1_ is generated by random selection, and a second-generation population *Q*_2_ is generated by a genetic operator. Subsequently, during the population iteration process, the offspring population *Q_t_* generated in generation t is merged together with the parent population P to form a new population of population size 2*N*. Then, the population *R_t_* is non-dominated sorted to find a series of non-dominated sets *Z*_1_, and the crowding degree of each individual is calculated. Since the individuals of both parent and child generations are contained in population *R_t_*, the non-dominated set *Z*_1_ after non-dominated sorting contains the best set of individuals in the whole population *R*_1_, so *Z*_1_ is put into the new parent population *P*_*t*+__1_ first. If at this point the size of the population is smaller than *N*, then it is necessary to continue to fill *P*_*t*+__1_ with the next level of non-dominated set *Z*_2_ until the size of the population exceeds *N* when the non-dominated set *Z_n_* is added, then, individuals are extracted using the crowding comparison operator for each individual in *Z_n_* to bring the size of the population *P*_*t*+*n*_ up to *N*. Genetic operators such as selection, crossover, and mutation are then used to generate new offspring populations *Q*_*t*+__1_ in order to prevent the results from not converging due to the infinite size of some parameters in the optimization process, and at the same time to avoid the structure not being processed or processing difficulties due to some parameters being too small. According to the results of the above analysis, the boundary of the optimization parameters is divided, and the boundary conditions are shown in Equation (31):(31)$$\left\{\begin{array}{c} 700\;\leq \;2a\;\leq \;1000\\ 00\;\leq \;2b\;\leq \;1000\\ 50\;\leq {\;h}_{d}\;\leq \;95\\ 200\;\leq {\;w}_{c}\;\leq \;400 \end{array}\right.$$

The weighted model trained by BP is imported into NSGA-II as the functional relationship, and the Pareto solution set obtained after 1000 iterations of the population is considered the optimal solution set. Each Pareto solution set contains the optimal points with their corresponding input parameters 2a, 2b, h_d_, and w_c_, as well as the optimized F and S. This process yields the optimal structure of the resonant pressure sensor and the highest sensitivity within the 10 MPa range.

#### 4.4.2. Optimization Results

The trained BP neural network algorithm accurately predicts two output variables, the interference frequency and operating frequency, based on four input variables: the diaphragm length, width, and height, as well as the width of the connecting beam. The optimization solution set obtained through the NSGA-II algorithm is depicted in Figure 12, where the horizontal axis represents the resonant frequency S of the structure at the full scale of 10 MPa, and the vertical axis represents the interference mode frequency F. Points below point C satisfy the requirement that the operating frequency is less than the interference frequency. Among them, point A represents the structure with the highest operating frequency under the condition of the operating frequency being lower than the interference frequency. According to Equation (24), the highest operating frequency corresponds to the highest sensitivity; thus, point A exhibits the highest sensitivity. However, at higher initial frequencies and tens of thousands of quality factors [17], in order to avoid the decrease in sensor measurement accuracy caused by the interference frequency approaching the operating frequency, a large frequency difference needs to be maintained between the two. Zhang F et al. [18] reported an initial frequency of 37,431 Hz for a sensor, with a difference of 5000 Hz from the interference frequency, thus avoiding the influence of interference modes. Therefore, we select point B as the optimal solution, with a frequency difference of 15,558 Hz between the interference frequency and the operating frequency, effectively avoiding the interference mode. The parameters at this point include a diaphragm width 2a of 713 μm, a height 2b of 999 μm, a thickness h_d_ of 50 μm, and a connecting beam width w_c_ of 234 μm. The resonant frequency at full scale is 163,751 Hz, and the interference frequency is 178,817 Hz. The optimization results were validated using the commercial software COMSOL, showing resonant and interference frequencies of 162,479 Hz and 178,493 Hz, respectively, which are in good agreement with the results of the NSGA-II multi-objective optimization. The specific structural parameters are listed in Table 2.

Within the full range of 10 MPa, the maximum internal stress of the resonator is 2.5 × 10^8^ Pa. After optimization, within the 10 MPa range, the relationship between the operating mode (Mode 5) and other modes is depicted in Figure 13. The frequency difference between the operating mode and the nearest interference mode within the range is greater than 15 kHz, effectively eliminating the influence of the interference modes. The sensitivity of the sensor within the operating range of 1–10 MPa is calculated to be 4.23 Hz/kPa, with a linearity of 0.9984 and a linear influence factor of 38.07. Table 3 lists the resonant pressure sensors proposed in recent years. Among them, the structure proposed by Y Lu and PC Yan has a sensitivity greater than 10 Hz/kPa, but its range is only around 1 MPa, resulting in a linear influence factor of around 10. The structures proposed by Y Jie have wide measurement ranges of 30 MPa and 50 MPa, but their sensitivities are only 0.4 and 0.066, resulting in linear influence factors of 12 and 3, respectively. The sensor structure designed in this study demonstrates good compatibility between the range and sensitivity, with a linearity impact factor significantly higher than that of most traditional resonant pressure sensors reported to date.

## 5. Conclusions

Traditional resonant pressure sensors often struggle to achieve both a wide range and high sensitivity simultaneously, limiting their applications in scenarios requiring both. To address this issue, we propose a novel sensor structure. By vertically distributing the resonant beam and pressure-sensitive diaphragm, we enhance the overall stability of the resonant structure and the equivalent stiffness of the diaphragm under high pressure. This effectively resolves the problem of reduced accuracy due to excessive deformation of the sensitive diaphragm, achieving good compatibility between the range and sensitivity. To mitigate the impact of parasitic modes during sensor operation, we present a multi-objective optimization design approach for resonant pressure sensors based on BP and NSGA-II algorithms. This approach addresses modal interference while optimizing the structural sensitivity. Validation of the optimization results using commercial software demonstrates a frequency optimization error of 0.7768% and an interference frequency optimization error of 0.1812%, confirming the reliability of the optimization approach. The optimized resonant structure achieves a high sensitivity of 4.23 Hz/kPa and a linear range of 1–10 MPa. The design strategy of the new structure provides guidance for the design of resonant pressure sensors with a wide range and high sensitivity, effectively addressing the compatibility issue between the sensitivity and range. The multi-objective optimization algorithm proposed for resonant pressure sensors can also be applied to optimize multiple parameters in other MEMS sensor structures, thereby improving the structural performance.

## Figures and Tables

**Figure 1 micromachines-15-00509-f001:**
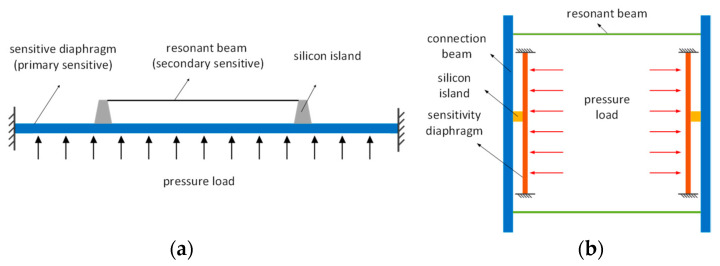
(**a**) Schematic diagram illustrating the working principle of the traditional beam–diaphragm structure; (**b**) schematic design of the new structure.

**Figure 2 micromachines-15-00509-f002:**
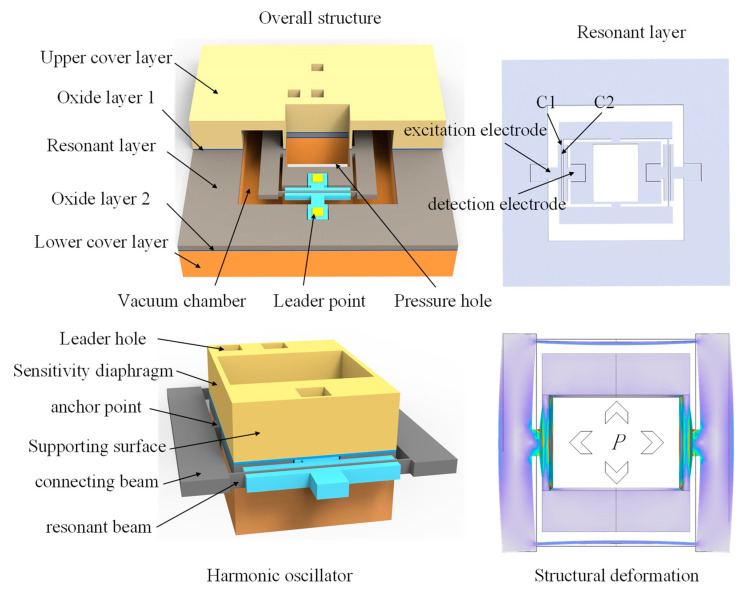
Schematic diagram of the vertical distribution resonator.

**Figure 3 micromachines-15-00509-f003:**
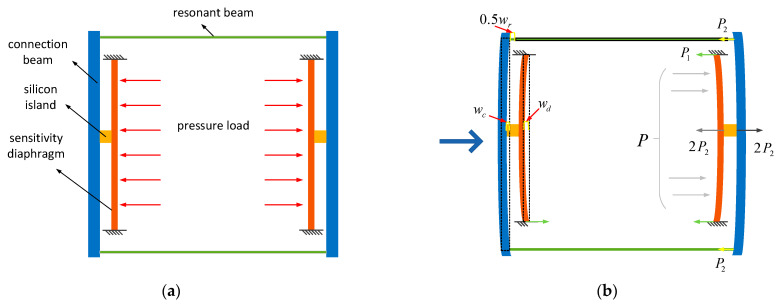
(**a**) Structural schematic diagram. (**b**) Stress and deformation analysis.

**Figure 4 micromachines-15-00509-f004:**
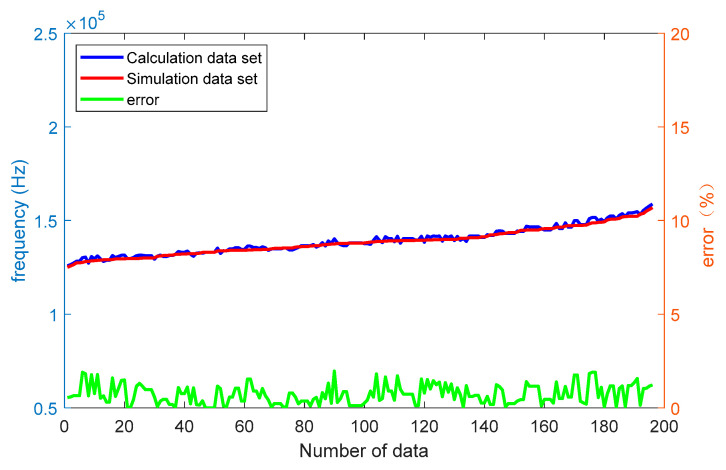
Comparison between mathematical model and simulation model.

**Figure 5 micromachines-15-00509-f005:**
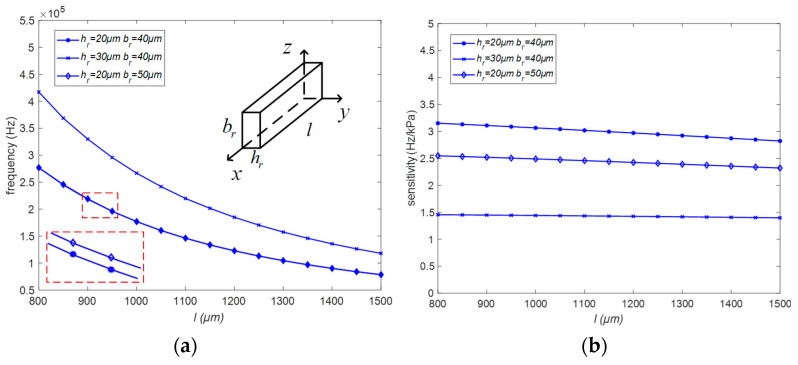
The influence of resonant beam parameters on the natural frequency and sensitivity: (**a**) Relationship between the parameters of the harmonic beam and the intrinsic frequency; (**b**) Trend of the influence of the variation of the parameters of the harmonic beam on the sensitivity.

**Figure 6 micromachines-15-00509-f006:**
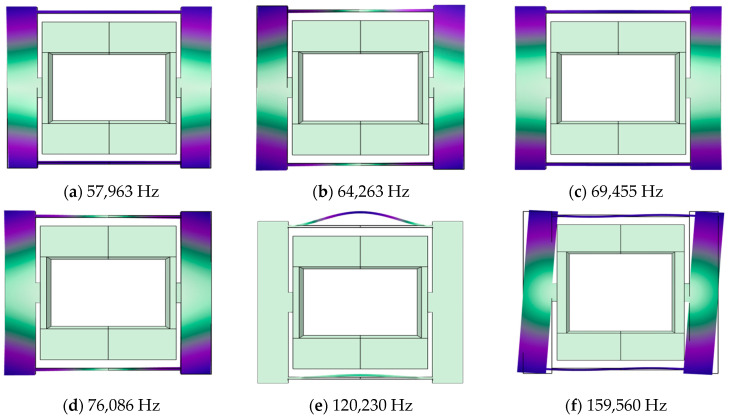
Dynamic analysis of the resonator: (**a**) the first mode shape; (**b**) the second mode shape; (**c**) the third mode shape; (**d**) the fourth mode shape; (**e**) the fifth mode shape; (**f**) the sixth mode shape, with the fifth mode as the operational mode and the sixth mode as the disturbance mode.

**Figure 7 micromachines-15-00509-f007:**
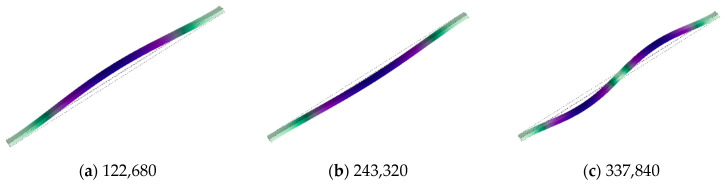
Vibration modes of the resonant beam: (**a**) the first mode shape; (**b**) the second mode shape; (**c**) the third mode shape.

**Figure 8 micromachines-15-00509-f008:**
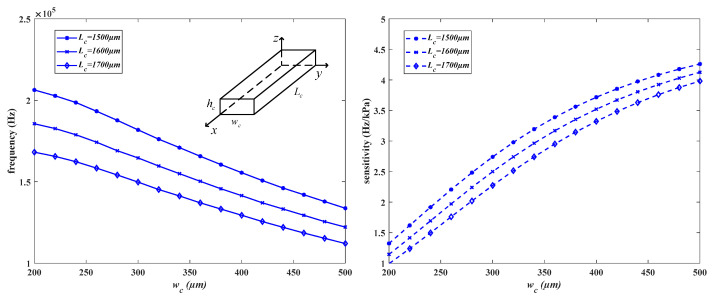
The impact of parameters on the disturbance frequency and sensitivity.

**Figure 9 micromachines-15-00509-f009:**
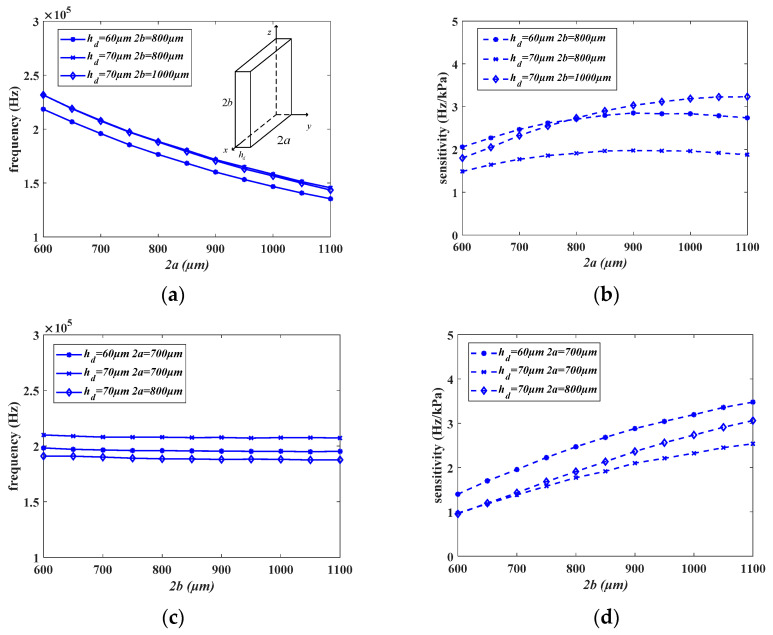
The relationship between diaphragm parameters and disturbance frequency and sensitivity. (**a**) Depicts the relationship between the width and disturbance frequency. (**b**) Illustrates the relationship between the width and sensitivity. (**c**,**d**) Show the influence of the height, while (**e**,**f**) demonstrate the effect of the thickness.

**Figure 10 micromachines-15-00509-f010:**
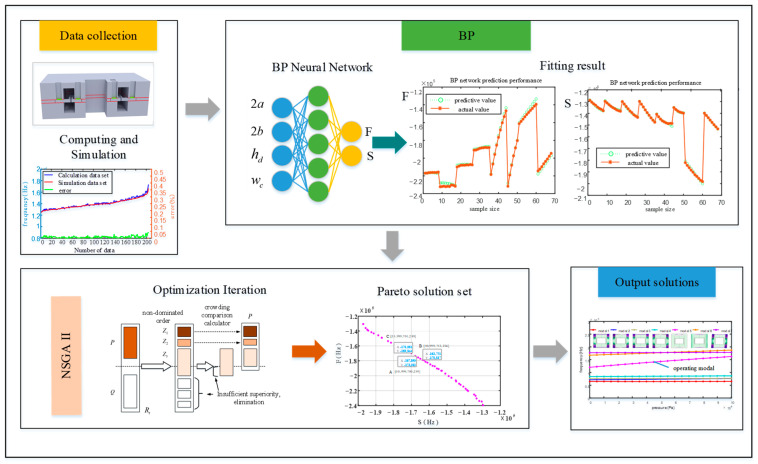
Optimization process.

**Figure 11 micromachines-15-00509-f011:**
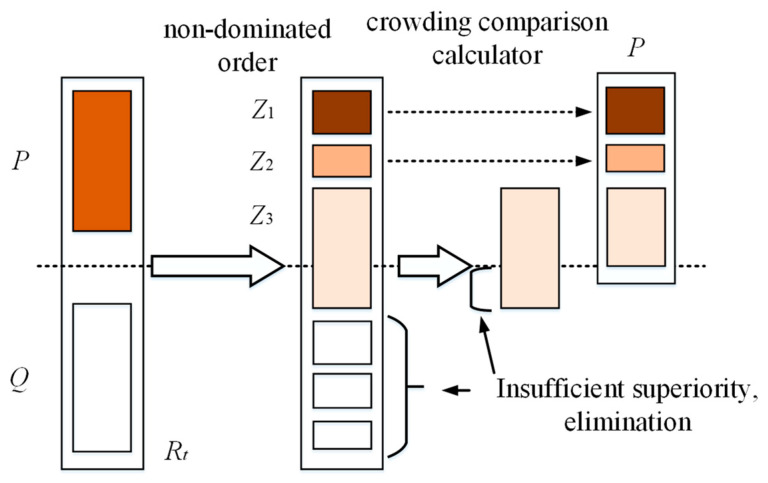
Elite strategy selection in NSGA-II algorithm.

**Figure 12 micromachines-15-00509-f012:**
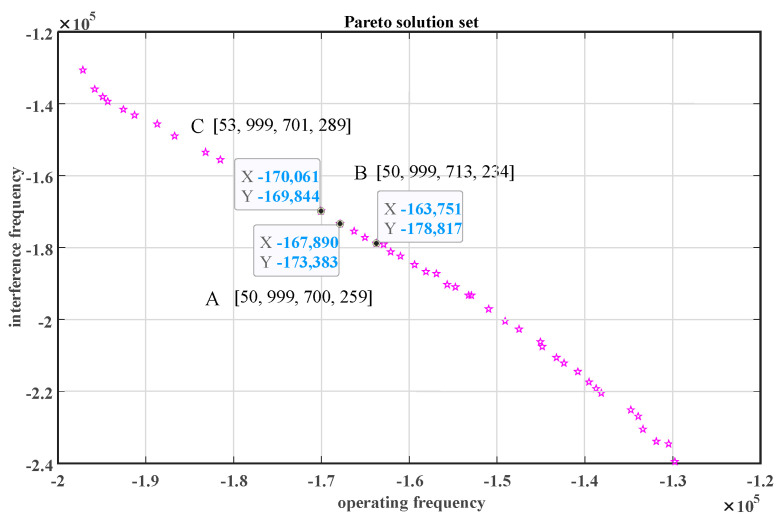
Pareto optimal solution set.

**Figure 13 micromachines-15-00509-f013:**
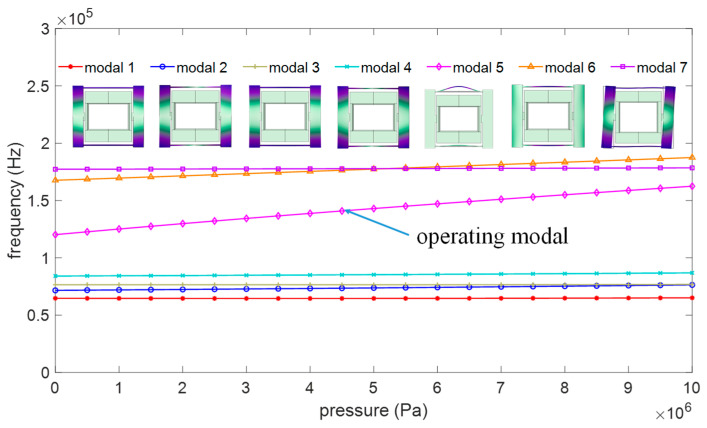
The relationship between resonant modes.

**Table 1 micromachines-15-00509-t001:** BP network training set.

h_d_/μm	2b/μm	2a/μm	w_c_/μm	S/Hz	F/Hz
50	800	600	230	−145,745.2954	−206,431.3251
50	800	600	240	−147,728.3998	−203,625.0286
60	800	700	250	−144,926.1623	−195,884.3529
65	800	700	250	−140,845.8125	−202,368.2791
70	800	700	250	−137,959.511	−207,989.4299
75	800	700	250	−135,335.4896	−213,111.4485
70	750	700	250	−136,039.2568	−207,997.5868
70	800	700	250	−137,959.511	−207,989.4299
70	850	700	250	−139,393.7201	−207,660.0328
60	800	650	250	−142,946.4384	−206,745.8811
60	800	700	250	−144,926.1623	−195,884.3529
60	800	750	250	−146,436.1059	−185,449.8793
60	800	800	250	−147,310.5528	−176,572.7048
60	800	850	250	−148,220.33	−168,312.7138
…	…	…	…	…	…

**Table 2 micromachines-15-00509-t002:** Key parameters of optimized structure.

Name	Definition of Parameters	Range (μm)
2a	Width of diaphragm	713
2b	Height of diaphragm	999
h_d_	Thickness of diaphragm	50
w_c_	Width of connect beam	234
L_c_	Length of connect beam	1626
h_c_	Height of connect beam	40
l	Length of resonant beam	1200
b_r_	Height of resonant beam	40
h_r_	Width of resonant beam	20
S	Resonant frequency	162,479 Hz
F	Interference frequency	178,493 Hz

**Table 3 micromachines-15-00509-t003:** Comparison of optimized structures.

Time	Author	Range (MPa)	Sensitivity (Hz/kPa)	Linear Influence Factor	Linearity
2019	Lu Y [8]	0.02–1	13.1	12.838	0.9999
2019	Yan P [7]	0.01–1	11.89	11.6522	0.9999
2020	Xiang C [19]	0.02–2	3.15	6.237	0.9999
2021	Xiang C [3]	0.02–7	2.26	18.0348	0.9998
2021	X Han [9]	0–0.2	35.5	7.01	0.9999
2022	Yu J [5]	0.11–30	0.428	12.84	0.9999
2022	Yu J [6]	0.11–50	0.066	3.3	0.9999
This structure		1–10	4.23	38.07	0.9984

## Data Availability

The original contributions presented in the study are included in the article, further inquiries can be directed to the corresponding author.

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
