# Peer review of "Design and Optimization of MEMS Resonant Pressure Sensors with Wide Range and High Sensitivity Based on BP and NSGA-II"

_micromachines, 2024, doi:10.3390/mi15040509_

Round 1
Reviewer 1 Report
Comments and Suggestions for Authors
1. BP and NSGA should be added to the abstract or introduction, only acronyms,confusing!
2. Two cases in Figure 5a and four cases in Figure 5b. Please remain consistent.
3. Explain why two algorithms are used for structure design. Purpose, features and innovations?
4. The parameter symbols in Figure 11 are blurred.
5. The information in Figure 10 is repeated in Figures 11, 12 and 13.
Comments on the Quality of English LanguagePlease revise grammatical errors in the manuscript.
Reviewer 2 Report
Comments and Suggestions for Authors
Dear Authors,
I am glad that I had the opportunity to familiarize myself with your important research in the field of design and creation of MEMS Resonant Pressure Sensors with Wide Range and High Sensitivity. I believe that you have created a development that is relevant for current use and has a good evidence base. But for now, I will have a few questions and suggestions for you, which I hope will help your future Readers understand your research in more detail:
1. Introduction. First paragraph. You provide excellent current examples from the field of functioning of pressure sensors based on the resonance method. But, for example, I work in the field of research on piezoresistive pressure sensors and it’s not entirely clear to me what linear sensing factor is? The nonlinearity error, like many other errors, must be presented as a percentage. Can you define linear sensing factor at the very beginning so that people from different areas of MEMS development can understand you?
2. Introduction. The second question also relates generally to the manner of writing the Introduction. You talk about the operation of resonant pressure sensors as a matter of course. But the same problems are also solved when developing pressure sensor chips using piezoresistive, capacitive or any other pressure conversion methods. Provide a brief confirmation of your choice of the resonance method for research.
3. Page 4. Line 133. Why is the ratio exactly 2 from deformation? For example, in piezoresistive transducers, in order to obtain the required low nonlinearity errors, the deviation of the diaphragm should be 1/3 of its thickness (beam thickness).
4. Page 7. Line 204-205. You indicated a range, but what type of pressure are you talking about? Differential, absolute or gauge?
5. Dear Authors, I will have general questions further, the description of which I did not see in your manuscript. Tell me please:
A. How many samples were included in the statistics and what is the technological spread between them in terms of output characteristics?
b. What housing design was the study conducted in? One of the two most important factors for measuring pressure sensor chips using any conversion method is the correct mechanical decoupling of the chip from the housing. Has the influence of residual mechanical stress been assessed?
With. Was the sample conditioning (temperature, pressure, etc.) carried out before taking measurements?
d. If we are not talking about third-party errors, for example, from the influence of temperature or time stability, then at least tell us about the mechanical parameters: what are the errors in mechanical hysteresis and repeatability?
Kind regards,
Reviewer
Round 2
Reviewer 1 Report
Comments and Suggestions for Authors
1. Curves can be observed in Figure 5a for only 2 cases. Need to zoom in?
Comments on the Quality of English Language
Minor editing of English language required.
Reviewer 2 Report
Comments and Suggestions for Authors
Dear Authors,
Thank you for your detailed response. I got all necessary answers so I can recommend to publish it in this form.
I wish you good luck for all your future research!
Kind regards,
Reviewer
